# Association between Serum GDF-15 and Cognitive Dysfunction in Hemodialysis Patients

**DOI:** 10.3390/biomedicines12020358

**Published:** 2024-02-03

**Authors:** Hae Ri Kim, Moo Jun Kim, Jae Wan Jeon, Young Rok Ham, Ki Ryang Na, Hyerim Park, Jwa-Jin Kim, Dae Eun Choi

**Affiliations:** 1Department of Nephrology, Chungnam National University Sejong Hospital, Sejong 30099, Republic of Korea; yo0118@cnuh.co.kr (H.R.K.); kimmoojun@cnuh.co.kr (M.J.K.); jeonjwan@cnuh.co.kr (J.W.J.); 2Department of Nephrology, Chungnam National University Hospital, Daejeon 35015, Republic of Korea; youngrok01@cnuh.co.kr (Y.R.H.); drngr@cnu.ac.kr (K.R.N.); kjj4827@gmail.com (J.-J.K.); 3Department of Medical Science, Medical School, Chungnam National University, Daejeon 35015, Republic of Korea; hye05240@gmail.com

**Keywords:** hemodialysis, GDF-15, cognitive dysfunction

## Abstract

Cognitive dysfunction is more frequent in end-stage renal disease (ESRD) patients undergoing hemodialysis compared with the healthy population, emphasizing the need for early detection. Interest in serum markers that reflect cognitive function has recently increased. Elevated serum growth differentiation factor 15 (GDF-15) levels are known to be associated with an increased risk of decreased renal function and cognitive dysfunction. This study investigated the relationship between GDF-15 and cognitive dysfunction in hemodialysis patients using a retrospective analysis of 92 individuals aged ≥ 18 years. Cognitive function was assessed using the Korean version of the Mini-Mental Status Examination (K-MMSE), categorizing patients into normal (≥24 points) and cognitive dysfunction (<24 points). As a result, serum GDF-15 concentrations were at significantly higher levels in the cognitive dysfunction group (7500.42 pg/mL, *p* = 0.001). Logistic regression indicated an increased risk of K-MMSE scores < 24 points when serum GDF-15 exceeded 5408.33 pg/mL. After indoxyl sulfate exposure in HT22 cells, HT22 cells survival was decreased and GDF-15 expression in HT22 cells was increased. Similarly, exposure to indoxyl sulfate in mouse brain tissue resulted in an increased expression of GDF-15. This study highlights the potential of serum GDF-15 as a marker for cognitive dysfunction in hemodialysis patients, offering a valuable screening tool. Serum GDF-15 is related to cognitive dysfunction in hemodialysis patients and may be helpful in screening for cognitive dysfunction in hemodialysis patients.

## 1. Introduction

As renal function decreases, the brain undergoes various changes. Chronic kidney disease (CKD) increases brain atrophy, a reduction in gray matter volume, and the disease burden of white matter [1,2,3]. In particular, the decreased volume of gray matter in hemodialysis patients causes changes in functional connectivity with other brain regions and is associated with functional brain defects [4]. Furthermore, as the enlarged perivascular spaces increase, the incidence of cerebral small vessel disease also increases [5]. Cerebral oxygenation is significantly reduced in CKD [6]. If brain oxygenation becomes unstable during hemodialysis, these hemodynamic changes may cause brain hypoxia and cognitive dysfunction [7].

Cognitive dysfunction is characterized by an impairment in specific cognitive domains caused by neurologic diseases or other systemic diseases. Cognitive function is evaluated by assessing cognitive domains, such as language, memory, attention, and executive functions [8]. Cognitive dysfunction is more frequent in end-stage renal disease (ESRD) patients undergoing hemodialysis compared with the healthy population [9].

Early suspicion and diagnosis of cognitive dysfunction are crucial because cognitive dysfunction is associated with a decreased quality of life as well as decreased drug adherence, increased mortality, and increased care costs [10,11,12,13].

Interest in serum markers that reflect cognitive function has recently increased. In previous studies, factors, such as total tau, amyloid α42, high-sensitivity C-reactive protein (CRP), and GDF-15, have been suggested to be serum markers for cognitive dysfunction [14,15]. Among these, serum GDF-15, a member of the transforming growth factor-cytokine superfamily, has an overall anti-inflammatory effect [16]. GDF-15 is thought to be elevated in inflammatory responses associated with cardiovascular disease, cancer, insulin resistance, and obesity [17]. A high serum GDF-15 level is known to be associated with a rapid decline in renal function as well as decreased renal function [18]. In a few studies, brain imaging revealed changes associated with brain atrophy in a group with high serum GDF-15 levels, and the risk of cognitive dysfunction was reportedly increased [19]. In addition, GDF-15 has been reported as a prognostic marker of mortality in hemodialysis patients [20,21,22].

Although there have been several studies on the relationship between GDF15 and mortality in hemodialysis patients, very few studies have evaluated serum GDF-15 as a marker for screening cognitive dysfunction in hemodialysis patients. This study investigated the relationship between cognitive dysfunction and serum GDF-15 levels in hemodialysis patients. In this study, our objective was to evaluate whether exposure to uremic toxins, observed in hemodialysis patients with an inclination towards elevated GDF-15 levels and cognitive dysfunction, would result in comparable outcomes in terms of GDF-15 expression in brain cells and tissues, as well as the survival of brain cells. The relationship between the expression of GDF-15 in mice and HT22 brain cells under conditions of increased uremia was investigated to determine whether uremic toxins were associated with an increase in brain GDF-15 levels.

In this study, we aimed to evaluate the potential of serum GDF-15 as a screening marker for cognitive decline in patients undergoing hemodialysis.

## 2. Materials and Methods

### 2.1. Study Population

This retrospective study analyzed data from hemodialysis patients treated at Chungnam National University Hospital, Daejeon, Republic of Korea, from January 2017 to June 2020. Ninety-two patients with end-stage renal disease on maintenance hemodialysis who underwent the Korean Mini-Mental Status Examination (K-MMSE) within 1 month of blood sample collection were included in the study population. The patients included in this study maintained hemodialysis three times a week, with each session lasting four hours. Patients diagnosed with dementia and taking medication were excluded. Patients diagnosed with depression or those taking medication for depression were excluded. This study conformed to the Declaration of Helsinki and was approved by the Ethics Committee of Chungnam National University Hospital (19 February 2021; Institutional Review Board approval no. CNUSH 2021-02-007).

### 2.2. Cognitive Function Assessment

The K-MMSE was used as a tool to evaluate cognitive dysfunction. The K-MMSE comprises seven components, encompassing spatial orientation, memory registration, memory recall, temporal orientation, language function, comprehension and judgment, and attention and calculation, and has a maximum possible score of 30 [23]. A score of 20–23 points is considered suspected dementia, and a score of ≤19 points is considered determined dementia. The cutoff point for cognitive dysfunction is a score of 23 [24]. In this study, the K-MMSE results for patients aged > 60 years were corrected according to their educational background. Thus, for uneducated patients, one point was added for temporal orientation, two for attention and calculation, and one for language function. However, the corrections were made so that each question did not exceed the maximum score. Furthermore, cognitive function in this study was classified as normal if the K-MMSE score was ≥24 or as cognitive dysfunction if the score was <24. For comparing the characteristics of the patient group based on cognitive function, the hemodialysis patients were divided into a normal cognitive function group (group A) and a cognitive dysfunction group (group B) based on the K-MMSE score.

### 2.3. Clinical Parameters

Patients’ clinical data were obtained from electronic medical records. Data collected included age, sex, body mass index, medical history, date of hemodialysis initiation, the period from hemodialysis initiation to the date of the K-MMSE, and education level. Medical history included hypertension, diabetes mellitus (DM), cerebral infarction, ischemic heart disease, and the causative disease of end-stage renal disease (ESRD). Blood sampling was performed immediately before hemodialysis. Laboratory data for serum GDF-15, serum creatinine, albumin, blood urea nitrogen (BUN), total protein, total cholesterol, total calcium, serum phosphorus, sodium, potassium, total CO_2_, and CRP were collected.

### 2.4. Measurement of Serum GDF-15 in Humans

The biospecimens and data used for further analysis on GDF-15 were provided by the Biobank of Chungnam National University Hospital, a member of the Korea Biobank Network. Blood samples for the measurement of serum GDF-15 were collected before dialysis on the day of hemodialysis. The samples were centrifuged immediately after collection and stored at −80 °C prior to use. Serum GDF-15 concentrations were determined in duplicate using a quantitative enzyme-linked immunosorbent assay kit (Human GDF-15 Quantikine ELISA Kit, R&D Systems, Minneapolis, MN, USA). All samples were analyzed in duplicate and measured according to the manufacturer’s instructions. In this study, the serum GDF-15 level was used as the average of the two replicates.

### 2.5. Mice and Drugs

To confirm that similar results were obtained in the in vitro and in vivo experiments regarding the analysis of GDF-15 measurement and cognitive decline in hemodialysis patients, experiments were also conducted in mice.

All animal experiments were performed using 10-week-old male C57BL/6 mice weighing 22–25 g each (Damul Science, Daejeon, Republic of Korea). The mice were bred under managed conditions at a controlled temperature of 22–23 °C under 50–55% humidity with a 12/12 h light/dark cycle. Animal care and all experiments were performed according to the institutional guidelines of the Korean Research Institute of Biotechnology and Bioscience, and the mice were cared for according to the protocol approved by the Chungnam National University Institutional Animal Care and Use Committee (202012A-CU-161). The mice were divided into two groups: sham (*n* = 5) and ischemia–reperfusion injury (IRI) (*n* = 6). All surgeries were performed under anesthesia with ketamine (60 mg/kg) and xylazine (8 mg/kg), and all efforts were made to minimize suffering. IRI was performed as previously described [25]. Briefly, following an abdominal incision, both renal pedicles were bluntly clamped for 35 min to induce ischemia. The mouse’s body temperature was maintained at 31–33 °C during the procedure. The sham-treated control mice underwent a similar surgical procedure without clamping. At 72 h after the IRI or sham operation, mice brains were collected under anesthesia. The mice died due to bleeding from the excision site while under anesthesia.

### 2.6. Blood and Tissue Preparation

Blood samples were collected from the inferior vena cava at sacrifice under anesthesia and placed into prechilled Eppendorf tubes (4 °C). The serum was separated by centrifugation for 20 min at 4 °C. Then, serum aliquots were shock-frozen using liquid nitrogen and stored at −70 °C. The brain tissue samples were prepared as previously described [26]. The brain was perfused transcardially with 4% PFA in PBS, and the brain tissues were collected for Western blot analysis.

### 2.7. Immunoblots

The brain tissue from the sham and IRI mice was dissected and homogenized in lysis buffer. Then, the brain tissue lysates (15 μg) were separated by 10% or 15% sodium dodecyl sulfate–polyacrylamide gel electrophoresis and transferred to nitrocellulose membranes. The primary antibodies (1:1000) used to probe the blots were anti-GDF-15 (Abcam, Cambridge, CB2 OAX, UK) and anti-α-tubulin (Cell Signaling Technology, Inc., Beverly, MA, USA). The blots were then incubated for 2 h with anti-rabbit IgG-HRP-linked antibody (1:1000) and anti-mouse IgG-HRP-linked antibody (1:1000) (Cell Signaling Technology, Inc.) as secondary antibodies. The protein bands were visualized using a chemiluminescence detection kit (Thermo Scientific, South Logan, UT, USA). The optical density of the proteins was determined using Gel-Pro Analyzer v3.1 software (Media Cybernetics, Silver Spring, MD, USA) to quantify the protein amount.

### 2.8. Cell Viability Assay

Brain cell injury was confirmed on the treatment of uremic toxin. Mouse hippocampal neuronal cell line HT22 was incubated with Dulbecco’s modified Eagle’s medium (DMEM; WELGENE, Gyeongsan-si, Republic of Korea) containing 10% fetal bovine serum (FBS; GenDEPOT, Katy, TX, USA) and 100 U/mL of penicillin–streptomycin (Gibco, Waltham, MA, USA) at 37 °C under 5% CO_2_. The MTT (3-(4,5-dimethylthiazol-2-yl)-2,5-diphenyltetrazolium bromide; Sigma-Aldrich, St. Louis, MO, USA) assay was used to measure cell viability via mitochondrial reductase. Briefly, HT22 cells were seeded into a 96-well plate at a density of 5 × 103 cells/well in DMEM containing 10% FBS and incubated for 24 h. Then, the culture medium was treated with indoxyl sulfate (a representative uremic toxin) at concentrations of 1, 5, and 10 mM, respectively, for 24 h. Next, MTT solution was added to each well to a final concentration of 0.5 mg/mL and incubated for 2 h at 37 °C. After removing the culture medium containing MTT, dimethyl sulfoxide was added, and the plate was incubated to dissolve the reduced formazan crystals at 37 °C for 1 h. Finally, the absorbance was measured at 540 nm.

### 2.9. Statistical Analysis

All statistical analyses were performed using SPSS version 26.0 (IBM Corp., Armonk, NY, USA). The *t*-test was used to compare continuous variables with a normal distribution, and the results were expressed as the mean and standard deviation. Categorical variables were compared using the chi-squared test. The correlation between K-MMSE scores and other parameters was analyzed using the Pearson correlation coefficient, and the association between serum GDF-15 and K-MMSE scores was examined with univariate and multivariate linear regression analyses. We used hierarchical regression analysis to assess the extent to which GDF-15 provides additional predictive value or validity in relation to age. Furthermore, the odds ratios and 95% confidence interval (CI) for the risk of acquiring an MMSE score of <24 were estimated with logistic regression analysis. A receiver operating characteristic (ROC) curve of the GDF-15 results was plotted to determine its sensitivity and specificity in predicting cognitive dysfunction. Using the ROC curve, a cutoff value of GDF-15 that is indicative of cognitive dysfunction (K-MMSE score < 24) was derived. Using the Youden index, the cutoff value for GDF-15 was set to 5408.33 pg/mL. Based on this cutoff value, a low serum GDF-15 level group (group C) and a high serum GDF-15 level group (group D) were made, and their characteristics were analyzed [27]. Data obtained from cells and mice tissues were analyzed using the Kruskal–Wallis non-parametric ANOVA test with Dunn’s test for multiple comparisons. The null hypothesis (no difference) was rejected if the *p*-value was <0.05.

## 3. Results

### 3.1. Baseline Characteristics

The initial study included 95 patients. However, 3 patients had dementia and were excluded, so 92 patients were included in the final analysisBased on the K-MMSE score classification used in this study, group A (K-MMSE score ≥ 24) comprised 59 patients, and group B (K-MMSE score < 24) comprised 33 patients. The mean K-MMSE score was 26.41 ± 4.469 points in group A and 19.36 ± 3.560 points in group B, and the older age of group A was statistically significant. DM was the predominant cause of ESRD in both groups, but the rates of hypertension and glomerulonephritis differed. DM was the underlying disease in 66.1% and 60.6% and hypertension was the underlying disease in 74.6% and 87.9% of patients in groups A and B, respectively. Regarding educational background, 83% of patients in group A had received >7 years of education, whereas 48.5% of patients in group B had a low level of education (Table 1).

The mean serum GDF-15 level in the two groups was 4808.22 vs. 7500.42 pg/mL, which was statistically significant (*p* = 0.001). On comparing group A with group B, the mean hemoglobin, BUN, serum creatinine, and potassium levels in group A were found to be higher than in group B, but these differences were not statistically significant. On the other hand, the levels of total cholesterol and CRP in group B were not statistically significant compared with group A (Table 1).

### 3.2. The Serum GDF-15 Level Was Correlated with the K-MMSE Score

Serum GDF-15 (r = −0.337, *p* = 0.001), age (r = −0.435, *p* = 0.000), and CRP (r = −0.257, *p* = 0.014) were negatively correlated with the K-MMSE score and serum albumin (r = 0.259, *p* = 0.013). Serum creatinine (r = 0.258, *p* = 0.013), potassium (r = 0.257, *p* = 0.013), and phosphorus (r = 0.348, *p* = 0.001) were positively correlated with the K-MMSE score (Table 2). A scatterplot also revealed a correlation between the K-MMSE score and GDF-15; the coefficient of determination was 0.114 (Figure 1 and Table 2).

The association between GDF-15 and the K-MMSE score was confirmed. A multivariate linear regression analysis was performed, where the variables included GDF-15, age, serum albumin, serum creatinine, potassium, phosphorus, and CRP levels. GDF-15 was confirmed as a significant independent factor with a negative correlation (*p* = 0.007). Similarly, age also showed a significant negative correlation (*p* = 0.001) (Table 3).

The extent to which GDF-15 provides additional predictive value or validity in relation to age was assessed using hierarchical regression analysis (Table 4). In Model 1, the variables included serum creatinine, albumin, potassium, and phosphorus. Age was introduced in Model 2, resulting in an R^2^ change of 0.108. Subsequently, in Model 3, GDF-15 was added as a variable, leading to a substantial increase in the R^2^ change to 0.61. These results suggest that the inclusion of GDF-15 in Model 3 likely enhanced incremental validity, indicating an improved predictive capacity in relation to the studied variables.

### 3.3. Elevated Serum GDF-15 Indicates Decreased Cognitive Function

An ROC curve was plotted to determine the area under the ROC curve (AUC) of the GDF-15 level and to identify an optimal cutoff value that predicated cognitive dysfunction. The GDF-15 AUC was 0.701 ± 0.058 (95% CI, 0.588–0.0.813; *p* = 0.01). A serum GDF-15 level > 5408.332 pg/mL exhibited 63.6% sensitivity and 64.4% specificity when distinguishing between normal and mild to severe cognitive impairment (Figure 2), and this was set as the cutoff value.

The patients were classified into either group A (serum GDF-15 ≤ 5408.332 pg/mL) or group B (serum GDF-15 > 5408.332 pg/mL) based on the determined GDF-15 cutoff value. Each variable was adjusted to the hazard ratio (HR) at GDF-15 > 5408.332 pg/mL to screen for cognitive dysfunction. Using logistic regression, the HR was derived by sequentially applying the variables that showed correlation across the three models. In particular, serum creatinine, serum albumin, potassium, and phosphorus applied in Model 1 showed the highest HR (HR: 3.596, *p* = 0.008). When adjusted for age, the HR was 3.089, which was also statistically significant (*p* = 0.034) (Table 5).

The characteristics of groups C and D were compared. The mean K-MMSE score was 22.21 ± 6.261 in group D, which was significantly lower than that of group A (25.28 ± 4.02) (*p* = 0.006). Regarding age, although group D was older than group C, this was not statistically significant. The proportion of patients with DM and hypertension was higher in group D, but it was not statistically supported. No significant difference was observed between the two groups except for the GDF-15 levels and K-MMSE scores (Table 6).

### 3.4. Brain Experiments of GDF-15 and Uremic Mice

Both BUN and serum creatinine were significantly elevated in mice with renal ischemia–reperfusion-induced azotemia compared with the sham mice (*p* < 0.05) (Figure 3). The expression of GDF-15 in the brain of azotemia-induced mice was significantly increased compared with the sham mice (Figure 4).

### 3.5. GDF-15 Expression in the Brain Tissue and Cells of Uremic Mice

When HT22 cells were treated with indoxyl sulfate, the survival rate decreased in a concentration-dependent manner (Figure 5). Moreover, indoxyl sulfate treatment increased the expression of GDF-15 in a concentration-dependent manner (Figure 6).

## 4. Discussion

This study confirmed the association between high levels of serum GDF-15 and cognitive dysfunction in ESRD on maintenance hemodialysis. Additionally, GDF-15 expression was revealed as increased in the brain tissue of uremic mice compared with normal mice.

Cognitive dysfunction is common in patients with CKD undergoing dialysis, and cognitive dysfunction has been reported in up to 38% of hemodialysis patients [10,28,29]. Additionally, cognitive dysfunction in patients with decreased renal function affects not only quality of life but also mortality [30]. Cognitive dysfunction in CKD is thought to be caused by vascular cognitive impairment, such as stroke and transient ischemic attack, and neurodegenerative processes, such as chronic hypertension, chronic cerebral inflammation, uremic toxins, and high α-amyloid levels. Malnutrition–inflammation–atherosclerosis syndrome and inflammation persisted in CKD patients undergoing dialysis, which may affect systemic blood vessels and organs [31,32,33].

The two most frequently used screening tests to evaluate cognitive function are the MMSE and the Montreal Cognitive Assessment [34,35]. In addition to using test scores, there have been several attempts to assess cognitive function using biochemical markers. The quantification of several serum markers related to cognitive function, such as total tau, amyloid α42, and high-sensitivity CRP, have been reported [14,36].

GDF-15 was reported to be associated with cognitive impairment and dementia in general populations [15,37]. In patients with CKD who are not undergoing dialysis, eGFR shows a tendency to decrease as GDF-15 levels increase [18,38]. Additionally, higher levels of serum GDF-15 have been reported to be associated with a rapid decline in renal function [39]. However, information regarding the relationship between cognitive function and GDF-15 in ESRD is lacking. Although GDF-15 levels are elevated in hemodialysis patients compared with patients with normal renal function, it is unclear whether the serum level of GDF-15 can be used as an indicator of cognitive dysfunction in hemodialysis patients [21].

In this study, using univariate and multivariate analyses, a statistically significant association was confirmed between high serum GDF-15 levels and a decline in cognitive function in hemodialysis patients. Furthermore, logistic regression analysis showed that the risk of cognitive dysfunction significantly increased by 2.912 times when the GDF-15 level was ≥5408.33 pg/mL in hemodialysis patients.

This study showed that age was independently correlated with the K-MMSE score. Multivariate linear regression results showed a negative correlation between age and the K-MMSE score. The tendency of the MMSE score to decrease with increasing age has been reported in the general population [40,41]. In this study, it was also surmised that age is an important factor in the MMSE results of hemodialysis patients. Although some studies have reported that GDF-15 increases with increased age in the healthy population, there was no significant relationship between GDF-15 and age in this study. These findings are presumed to result from decreased renal function, in addition to age-related changes, in hemodialysis patients [42,43]. In this study, the univariate analysis results showed that the K-MMSE score was associated with creatinine, albumin, phosphorus, potassium, and CRP. Creatinine is frequently used as an indicator of muscle mass in dialysis patients, and there is a possibility of sarcopenia in patients with low creatinine levels. The association between sarcopenia and cognitive dysfunction has been demonstrated in several studies [44,45,46].

Previous studies have reported on the relationship between malnutrition and cognitive dysfunction [47]. Malnourished patients are at increased risk of poor performance, and it is presumed that immobility is related to cognitive dysfunction. Additionally, hypophosphatemia has been reported as a potential marker of β-amyloid deposition associated with Alzheimer’s disease [48]. In this study, K-MMSE scores tended to decrease in patients with hypokalemia, hypophosphatemia, and hypoalbuminemia, which are commonly associated with malnutrition, showing a similar result to that obtained in the previous study.

In this study, we were unable to identify antecedent factors that could be considered in the relationship between aging and the elevation of GDF-15. Furthermore, the specific mechanism through which GDF-15 may induce cognitive dysfunction could not be definitively confirmed. Consequently, a clear conclusion regarding whether GDF-15 triggers cognitive dysfunction or increases in association with cognitive dysfunction could not be drawn. While GDF-15 may rise in situations such as aging and conditions similar to chronic kidney disease (CKD), the increase in GDF-15 concentrations when brain cells and tissues are exposed to uremic toxins (such as indoxyl sulfate) suggests that elevated levels of GDF-15 may indicate brain damage in the context of cognitive decline. Especially in patients undergoing hemodialysis, considering GDF-15 elevation beyond a certain threshold as a potential marker for screening or suspicion of cognitive dysfunction appears plausible.

There are several limitations in this study. First, this study was a retrospective study and had a small sample size, which limited the analysis of GDF-15 trends by age. When the cutoff value of serum GDF-15 was set at 5408.33 pg/mL in the ROC curve, it showed 63.6% sensitivity and 64.4% specificity in predicting cognitive dysfunction. The applicability of these findings to a broader population is limited because the sample size was too small to set an accurate cutoff value that could be used as a marker of cognitive function. It is expected that the prediction rate can be further improved if a relatively larger amount of data is analyzed in future studies. Second, the results were analyzed based on patients from a single country, and consideration for other races was not taken into account. It seems necessary to consider multiple races in future studies. Third, as this study was retrospective, various tools could not be used to evaluate cognitive function, and only the K-MMSE was utilized. In future research, it is believed that richer results will be obtained by using multiple cognitive function assessment tools in retrospective studies. Fourth, among the patients included in this study, there were no individuals diagnosed with depression related to cognitive impairment. However, as undiagnosed depression was not assessed, it is considered crucial for future studies to evaluate not only cognitive function but also depression.

## 5. Conclusions

A few studies have evaluated serum GDF-15 as a marker for screening cognitive dysfunction in hemodialysis patients. In this study, GDF-15 levels were observed to be elevated in the group of hemodialysis patients with cognitive decline. These findings were further confirmed in mice experiments, where similar results were demonstrated. Several cognitive function evaluation tests, including the Mini-Mental Status Examination (MMSE), should be performed by trained medical personnel as it is costly and time-consuming. In contrast, the measurement of serum GDF-15 is relatively easy. In the future, it is considered that elevated serum GDF-15 levels may be helpful as a potential marker for screening cognitive function in hemodialysis patients.

## Figures and Tables

**Figure 1 biomedicines-12-00358-f001:**
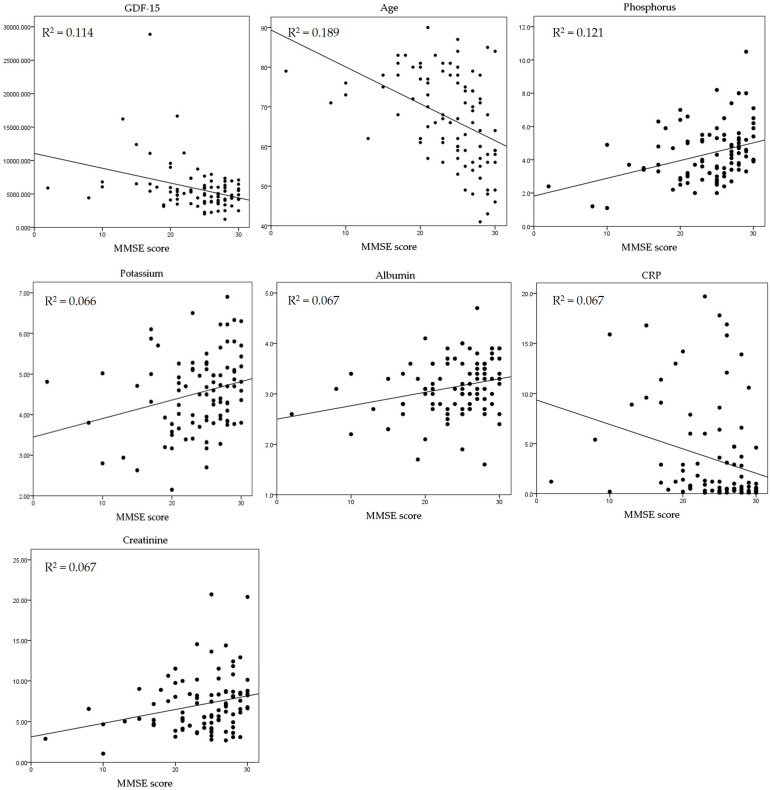
Correlation between K-MMSE score and the clinical parameters. GDF-15 = growth and differentiation factor 15; CRP = C-reactive protein.

**Figure 2 biomedicines-12-00358-f002:**
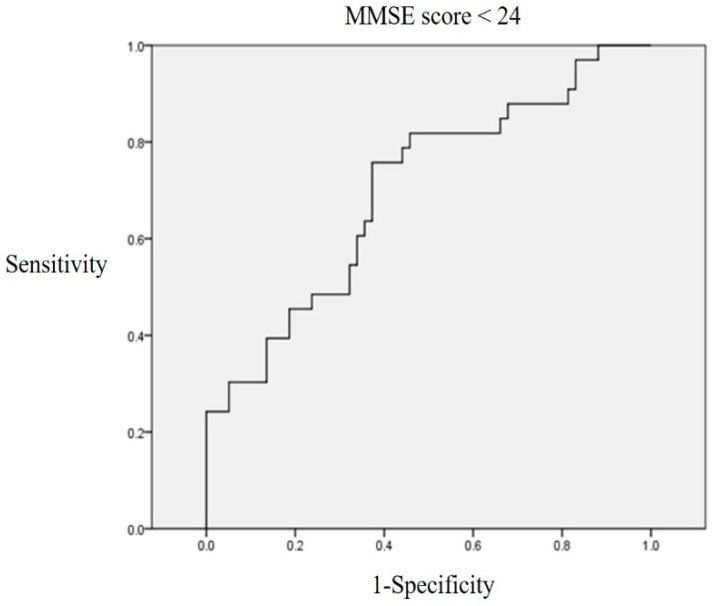
ROC curve linking the GDF-15 level to a K-MMSE score < 24. A serum GDF-15 level > 5408.33 pg/mL exhibited 63.6% sensitivity and 64.4% specificity when distinguishing between normal and mild to severe cognitive dysfunction.

**Figure 3 biomedicines-12-00358-f003:**
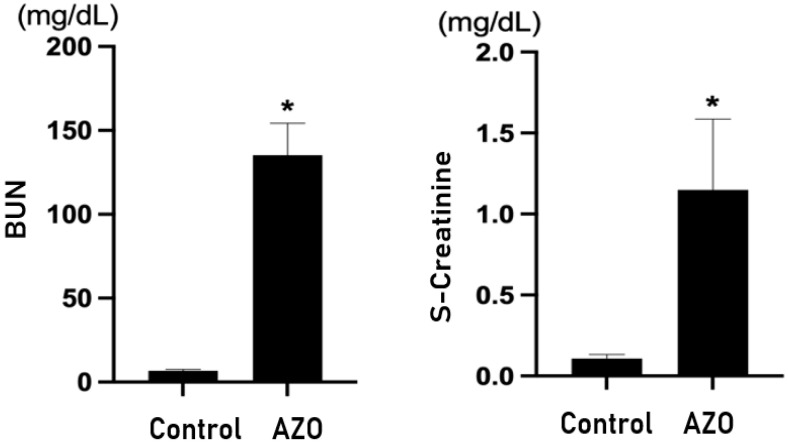
Renal function in an ischemia–reperfusion-induced renal injury (IRI) model. s-Creatinine = serum creatinine; Control = sham; AZO = renal ischemia–reperfusion-induced azotemia; BUN = blood urea nitrogen. * *p* value < 0.05, Control (sham) vs. AZO (IRI).

**Figure 4 biomedicines-12-00358-f004:**
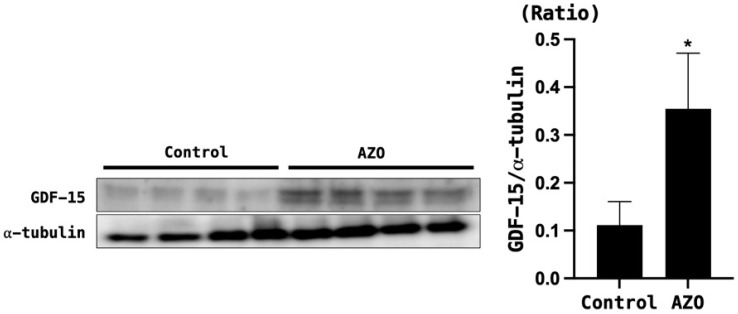
GDF-15 levels in the brain tissue of mice with azotemia induced by ischemia–reperfusion injury (IRI). α-tubulin was used as the control. Control = sham; AZO = IRI. * *p* < 0.05, Control (sham) vs. AZO (IRI model).

**Figure 5 biomedicines-12-00358-f005:**
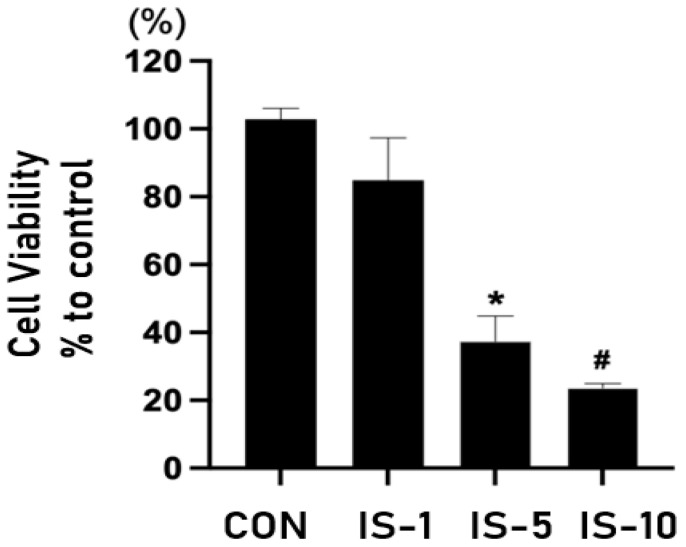
Survival of mouse hippocampal neuronal cell line HT22 following indoxyl sulfate treatment at concentrations of 1, 5, and 10 mM. IS = indoxyl sulfate. * *p* < 0.05, control vs. indoxyl sulfate (IS) 5 mM, # *p* < 0.05, control vs. IS 10 mM.

**Figure 6 biomedicines-12-00358-f006:**
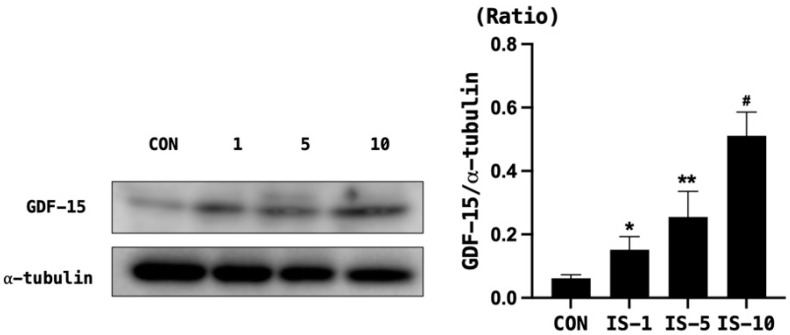
GDF-15 level in mouse hippocampal neuronal cell line HT22 following indoxyl sulfate treatment. GDF-15 expression increased in a concentration-dependent manner. α-tubulin was used as the control. Con, control, GDF-15 = growth and differentiation factor-15, IS = indoxyl sulfate. * *p* < 0.05, control vs. IS 5 mM, ** *p* < 0.05, control vs. IS 5 mM, # *p* < 0.05, control vs. IS 10 mM.

**Table 1 biomedicines-12-00358-t001:** Baseline patient characteristics according to the K-MMSE score (group A, score ≥ 24; group B, score < 24) (*N* = 92).

	Group A (*n* = 59)	Group B (*n* = 33)	*p*-Value
K-MMSE score	26.41 ± 4.469	19.36 ± 3.560	<0.01 *
Age (years) (Mean ± SD)	64.03 ± 11.675	72.76 ± 8.700	<0.01 *
Gender (*n*, %)			0.16 †
Male	38 (64.4%)	19 (57.6%)	
Female	21 (35.6%)	14 (42.4%)	
ESRD causes (*n*, %)			
DM	39 (66.1%)	20 (60.6%)	0.31 †
Hypertension	6 (10.2%)	7 (21.2%)	
Chronic GN	9 (15.3%)	4 (12.1%)	
PCKD	0 (0%)	1 (3.0%)	
Others	5 (8.5%)	1 (3.0%)	
Medical history (*n*, %)			
DM	40 (67.8%)	23 (69.7%)	0.85 †
Hypertension	44 (74.6%)	29 (87.9%)	0.13 †
IHD	12 (20.3%)	7 (21.2%)	0.92 †
CI	7 (11.9%)	5 (15.2%)	0.65 †
HD vintage (days) (Mean ± SD)	618.69 ± 1158.8	305.12 ± 380.7	0.14 *
Education duration (years)			<0.01 †
0	1 (1.7%)	6 (18.2%)	
1–6	9 (15.3%)	11 (33.3%)	
7–9	11 (18.6%)	5 (15.2%)	
10–12	26 (44.1%)	8 (24.2%)	
>12	12 (20.3%)	3 (9.1%)	
BMI (kg/m^2^) (Mean ± SD)	24.098 ± 3.730	22.930 ± 3.877	0.17 *
Laboratory test (Mean ± SD)			
GDF-15 (pg/mL)	4808.22 ± 1585.21	7500.42 ± 5080.78	<0.01 *
Hemoglobin (g/dL)	12.21 ± 21.49	9.43 ± 1.24	0.33 *
Total protein	6.14 ± 0.85	5.98 ± 0.72	0.34 *
Albumin	3.21 ± 0.54	3.00 ± 0.55	0.08 *
Total cholesterol	138.09 ± 47.92	148.08 ± 69.66	0.52 *
BUN	62.43 ± 36.08	58.42 ± 32.41	0.59 *
Serum creatinine	7.41 ± 3.75	6.62 ± 2.89	0.26 *
Total calcium	8.00 ± 0.99	7.93 ± 0.76	0.68 *
Phosphorus	4.55 ± 1.73	4.05 ± 1.45	0.14 *
Sodium	136.49 ± 4.27	135.99 ± 4.57	0.61 *
Potassium	4.68 ± 0.86	4.28 ± 1.06	0.07 *
Total CO_2_	20.88 ± 4.82	21.47 ± 3.20	0.50 *
CRP	2.81 ± 4.52	4.91 ± 5.76	0.06 *
B2-MG	21.54 ± 12.14	21.10 ± 11.27	0.87 *
Homocysteine	17.83 ± 10.01	17.14 ± 7.78	0.70 *

SD, standard deviation; K-MMSE, Korean Mini-Mental Status Examination; ESRD, end-stage renal disease; DM, diabetes mellitus; GN, glomerulonephritis; PCKD, polycystic kidney disease; IHD, ischemic heart disease; CI, cerebral infarction; HD, hemodialysis; BMI, body mass index; GDF 15, growth and differentiation factor 15; BUN, blood urea nitrogen; CRP, C-reactive protein; B2-MG, beta-2 microglobulin. * Student’s *t*-test. † Chi-square test.

**Table 2 biomedicines-12-00358-t002:** Correlation between the K-MMSE score and clinical parameters.

	r	*p*-Value
GDF-15	−0.337	<0.01
Age	−0.435	<0.01
BMI	0.149	0.16
Total protein	0.185	0.08
Albumin	0.259	0.01
Total cholesterol	0.030	0.80
BUN	0.191	0.07
Serum creatinine	0.258	0.01
Total calcium	0.042	0.69
Phosphorus	0.348	<0.01
Sodium	0.093	0.38
Potassium	0.257	0.01
Total CO_2_	−0.122	0.26
C-reactive protein	−0.257	0.01
B2-MG	−0.077	0.47
Homocysteine	0.121	0.26

BMI, body mass index; GDF-15, growth and differentiation factor 15; BUN, blood urea nitrogen; B2-MG, beta-2 macroglobulin.

**Table 3 biomedicines-12-00358-t003:** Association between GDF15 and the K-MMSE score (multivariate linear regression).

Variables	B	S.E	β	t	*p*-Value
(Constant)	32.166	6.216		5.175	0.000
GDF-15	0.000	0.000	−0.255	−2.780	0.007
Age	−0.178	0.050	−0.381	−3.536	0.001
Albumin	0.943	1.019	0.097	0.925	0.357
Cr	−0.102	0.237	−0.066	−0.431	0.668
Potassium	0.856	0.575	0.152	10.489	0.140
Phosphorus	0.100	0.418	0.031	0.240	0.811
CRP	−0.111	0.112	−0.105	−0.989	0.325

GDF-15, growth and differentiation factor-15; Cr, serum creatinine; CRP, C-reactive protein.

**Table 4 biomedicines-12-00358-t004:** Incremental contributions of the K-MMSE score (hierarchical regression).

	Model 1			Model 2			Model 3		
	B	SE	β	B	SE	β	B	SE	β
(Constant)	15.93	4.24		29.74	5.55		31.75	5.39	
Albumin	0.80	1.12	0.08	1.05	1.05	0.11	0.94	1.01	0.10
Cr	0.13	0.20	0.08	−0.07	0.19	−0.04	−0.08	0.19	−0.05
Phosphorus	0.64	0.43	0.20	0.17	0.42	0.05	0.11	0.41	0.03
Potassium	0.50	0.62	0.09	0.79	0.59	0.14	0.86	0.57	0.15
CRP	−0.15	0.12	−0.15	−0.15	0.11	−0.14	−0.11	0.11	−0.10
Age				−0.19	0.05	−0.40	−0.18	0.05	−0.38
GDF-15							0.00	0.00	−0.25
R^2^	0.17			0.28			0.34		
R^2^ change	0.172 *			0.108 **			0.61 **		

GDF-15, growth and differentiation factor-15; Cr, serum creatinine; CRP, C-reactive protein. * *p* < 0.05, ** *p* < 0.01. Model 1: adjusted for serum creatinine, albumin, potassium, and phosphorus. Model 2: adjusted for Model 1 + age. Model 3: adjusted for Model 2 + GDF-15.

**Table 5 biomedicines-12-00358-t005:** Assessment results of the risk of having a K-MMSE score of <24 in a logistic regression model (group A, score ≥ 24 and group B, score < 24).

	Group A	Group B	*p*-Value
Crude	Ref.	3.167 (1.153–6.485)	0.011
Model 1	Ref.	3.596 (1.280–7.066)	0.008
Model 2	Ref.	3.492 (1.251–6.665)	0.010
Model 3	Ref.	3.089 (1.128–4.471)	0.034

Crude: GDF-15 > 5408.33 pg/mL. Model 1: adjusted for GDF15 + serum creatinine, albumin, potassium, and phosphorus. Model 2: adjusted for Model 1 + C-reactive protein. Model 3: adjusted for Model 2 + age.

**Table 6 biomedicines-12-00358-t006:** Comparison of GDF-15 level between groups 3 and 4 (group C, ≤5408.33 pg/mL and group D, >5408.33 pg/mL).

	Group C (*n* = 50)	Group D (*n* = 42)	*p*-Value
K-MMSE score	25.28 ± 4.021	22.21 ± 6.261	0.006 *
Age (mean ± SD) (years)	65.36 ± 12.059	69.31 ± 10.426	0.096 *
Sex (*n*, %)			0.660 †
Male (*n*, %)	32 (64.0%)	25 (59.5%)	
Female (*n*, %)	18 (36.0%)	17 (40.5%)	
ESRD causes (*n*, %)			0.101 †
DM	30 (60%)	29 (69%)	
Hypertension	5 (10%)	8 (19%)	
GN	8 (16%)	5 (11.9%)	
PCKD	1 (2%)	0 (0%)	
Others	6 (12%)	0 (0%)	
Medical history (*n*, %)			
DM	30 (60%)	33 (78.6%)	0.056 †
Hypertension	36 (72%)	37 (88.1%)	0.057 †
IHD	9 (18%)	10 (23.8%)	0.493 †
CI	5 (10%)	7 (16.7%)	0.344 †
HD vintage (days) (mean ± SD)	508.08 ± 1124.32	501.195 ± 729.92	0.972 *
Education duration (years)			0.363 *
0	5 (10.0%)	2 (4.8%)	
1–6	7 (14.0%)	13 (31.0%)	
7–9	9 (18.0%)	7 (16.7%)	
10–12	20 (40.0%)	14 (33.35)	
>12	9 (18.0%)	6 (14.3%)	
BMI (kg/m^2^) (mean ± SD)	23.28 ± 3.80	24.16 ± 3.80	0.274 *
Laboratory test (mean ± SD)			
GDF-15 (pg/mL)	3988.53 ± 969.81	7899.34 ± 4219.57	0.000 *
Hemoglobin (g/dL)	65.36 ± 12.059	69.31± 10.426	0.096 *
Total protein	12.868 ± 23.301	9.248 ± 1.514	0.278 *
Albumin	6.056 ± 0.7478	6.107 ± 0.876	0.766 *
Total cholesterol	3.152 ± 0.559	3.119 ± 0.551	0.777 *
BUN	147.33 ± 60.784	134.84 ± 51.182	0.354 *
Serum creatinine	61.972 ± 37.557	59.819 ±31.329	0.765 *
Total calcium	9.688 ± 4.923	8.552 ± 3.594	0.205 *
Phosphorus	7.913 ± 0.992	8.052 ± 0.810	0.461 *
Sodium	4.446 ± 1.7385	4.274 ± 1.541	0.616 *
Potassium	136.622 ± 4.205	135.938 ± 4.564	0.461 *
Total CO_2_	4.4780 ± 0.823	4.6081 ± 1.093	0.527 *
C-reactive protein	21.067 ± 4.657	21.133 ± 3.835	0.942 *
B2-MG	3.033 ± 4.537	4.198 ± 5.635	0.286 *
Homocysteine	22.579 ± 13.283	20.005 ± 9.728	0.293 *

SD, standard deviation; K-MMSE, Korean Mini-Mental Status Examination; ESRD, end-stage renal disease; DM, diabetes mellitus; GN, glomerulonephritis; PCKD, polycystic kidney disease; IHD, ischemic heart disease; CI, cerebral infarction; HD, hemodialysis; BMI, body mass index; GDF-15, growth and differentiation factor 15; BUN, blood urea nitrogen; B2-MG, beta-2 microglobulin. * Student’s *t*-test. † Chi-square test.

## Data Availability

All relevant data are within this paper.

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
