# Peer review of "Association between Serum GDF-15 and Cognitive Dysfunction in Hemodialysis Patients"

_biomedicines, 2024, doi:10.3390/biomedicines12020358_

Round 1
Reviewer 1 Report
Comments and Suggestions for Authors
Minor revisions. I really enjoyed the study and provided significant outcomes for future research regarding haemodialysis patients and cognitive dysfunctions.
Two main concerns:
(1) I would run hierarchical analyses instead of multivariate linear regression. Why? it seems that age has a more important role than GDF-15. Hence, if you can calculate the incremental validity of GDF-15 to cognitive dysfunction then you may consider this serum as a mediator (a negative-impact one) between treatment and cognitive dysfunction.
(2) Consider reporting the both outcome and proceeding mice's study much better to be replicated by others.
Because of limitations (a retrospective study), the discussion should be written in a tentative style.
Y upload the file with suggestions and commentaries
Good job anyway

Author Response
(1) I would run hierarchical analyses instead of multivariate linear regression. Why? it seems that age has a more important role than GDF-15. Hence, if you can calculate the incremental validity of GDF-15 to cognitive dysfunction then you may consider this serum as a mediator (a negative-impact one) between treatment and cognitive dysfunction.
ANS) In this study, we did not definitively establish whether GDF-15 directly induces cognitive dysfunction in patients undergoing hemodialysis or if it is a factor that increases secondarily in cases of cognitive dysfunction. Hierarchical analyses were performed, revealing enhanced incremental validity upon introducing GDF-15 as a variable. Our analysis identified that in patients undergoing hemodialysis, an elevation in GDF-15 beyond a certain threshold is linked to an elevated risk of cognitive dysfunction. Given these results, the measurement of GDF-15 could be explored as a potential marker for screening cognitive dysfunction. Nevertheless, further analysis with a larger dataset is considered necessary for future investigations.
(2) Consider reporting the both outcome and proceeding mice's study much better to be replicated by others. Because of limitations (a retrospective study), the discussion should be written in a tentative style.
ANS) I have made revisions to the main text and addressed the comments in the PDF file. Please review the changes in PDF
Reviewer 2 Report
Comments and Suggestions for Authors
In this work, the authors confirmed the association between high levels of serum GDF-15 and cognitive dysfunction found both in vivo on humans (hemodialysis patients of a retrospective cohort) and on mice C57BL6 subjected to ischemia-reperfusion injury; and in vitro on a mouse hippocampal neuronal cell line HT22. The results are interesting mainly in relation to the role of GDF-15 as potential biomarker of cognitive dysfunction. The materials and methods are sound. There is a Limitations section.
I have some minor points to address:
1. in the abstract section, there are some typos errors: for instance, line 21 page 1 (serum instead of Serum); more importantly, the sentence from line 27 to line 28 should be eliminated because it seems to be redundant;
2. in the introduction section, the authors should better explain at the end of this section the originality and the added value of their work. It is important to underline what is known in literature and what the authors want to add as new and impactful prospective on this topic and on their work. Better to repeat this finding also in the conclusion section;
3. In the same section, from line 56 to line 58, the authors should report the papers related to these findings: “brain imaging showed changes associated with 56 brain atrophy in the group with high serum GDF-15 level, and the risk of cognitive dysfunction was reportedly increased.”
4. In the discussion section, the authors should be clearer and incisive mainly when (if pertinent) it is needed to compare their work with other authors/papers. To be clear when they referred to ESRD or CDK.
5. In the Limitations section, the authors could add that it is a limitation to use only the MMSE to assess the cognitive dysfunction.
6. Check for acronyms in the text: for instance, ESRD, CDK, HTN, MTT...
7. Check for English errors or typos, for instance “in another few studies…” (line 56 page 2); “in in vitro..” (line 115 page 3); serum creatinine (r =.258, P = 0.0213), il P=0.013 (line 211, page 6); “to be due to decreased renal function …” (line 342 page 12).
Comments on the Quality of English LanguageThere are some minor errors to address
Author Response
- in the abstract section, there are some typos errors: for instance, line 21 page 1 (serum instead of Serum); more importantly, the sentence from line 27 to line 28 should be eliminated because it seems to be redundant;
ANS) Modifications have been reflected in the text.
As a results, Serum serum GDF-15 concentrations were significantly higher levels in the cognitive dysfunction group (7500.42 pg/mL, P = 0.001). Logistic regression indicated an increased risk of K-MMSE scores < 24 points when serum GDF-15 exceeded 5408.33 pg/mL. After Uremic toxin exposure in HT22 cells (brain cell) in mouse model, cell survival was decreased and GDF-15 ex-pression was increased After Indoxyl sulfate exposure in HT22 cells, HT22 cells survival was de-creased and GDF-15 expression in HT22 cells was increased. Similarly, exposure to indoxyl sulfate in mouse brain tissue resulted in an increased expression of GDF-15.This study highlights the potential of serum GDF-15 as a marker for cognitive dysfunction in hemodialysis patients, offering a valuable screening tool. Serum GDF-15 is related to cognitive dysfunction in hemodialysis patients and may be helpful in screening for cognitive dysfunction in hemodialysis patients. Serum GDF-15 is related to cognitive dysfunction in hemodialysis patients and may be helpful in screening for cognitive dysfunction in hemodialysis patients.
- in the introduction section, the authors should better explain at the end of this section the originality and the added value of their work. It is important to underline what is known in literature and what the authors want to add as new and impactful prospective on this topic and on their work. Better to repeat this finding also in the conclusion section;
ANS) Additional modifications were made in the introduction and conclusion part.
*introduction part
However, there have been several studies on the relationship between GDF15 and mortality in hemodialysis patients, but very few studies have evaluated serum GDF-15 as a marker for screening cognitive dysfunction in hemodialysis patients. This study investigated the relationship between cognitive dysfunction and serum GDF-15 levels in hemodialysis patients. Additionally, the potential of GDF-15 as a marker related to cog-nitive dysfunction in patients undergoing hemodialysis was evaluated. The relationship between the expression of GDF-15 in mice and HT22 brain cells under conditions of in-creased uremia was investigated to determine whether uremic toxins were associated with an increase in brain GDF-15 levels.
In this study, we aimed to evaluate the potential of serum GDF-15 as a screening marker for cognitive decline in patients undergoing hemodialysis.
*conclusion part
A few studies have evaluated serum GDF-15 as a marker for screening cognitive dysfunction in hemodialysis patients. In this study, GDF-15 levels were observed to be elevated in the group of hemodialysis patients with cognitive decline. These findings were further confirmed in mice experiments, where similar results were demonstrated. Several cognitive function evaluation tests, including Mini-Mental Status Examination (MMSE), should be performed by trained medical personnel, and it is costly and time-consuming. In contrast, the measurement of serum GDF-15 measurement is rela-tively easy. The MMSE should be performed by trained medical personnel, and it is costly and time-consuming. In contrast, the measurement of serum GDF-15 measure-ment is relatively easy. In the future, it is considered that elevated serum GDF-15 levels may be helpful as a potential marker for screening cognitive function in hemodialysis patients.This study shows that serum GDF-15 levels may be helpful to screen for cogni-tive dysfunction in hemodialysis patients.
- In the same section, from line 56 to line 58, the authors should report the papers related to these findings: “brain imaging showed changes associated with 56 brain atrophy in the group with high serum GDF-15 level, and the risk of cognitive dysfunction was reportedly increased.”
ANS) Modifications have been reflected in the text.
In a few studies, brain imaging revealed changes associated with brain atrophy in the group with high serum GDF-15 levels, and the risk of cognitive dysfunction was re-portedly increased [19].
* reference : McGrath, E.R.; Himali, J.J.; Levy, D.; Conner, S.C.; DeCarli, C.; Pase, M.P.; Ninomiya, T.; Ohara, T.; Courchesne, P.; Satizabal, C.L.; et al. Growth Differentiation Factor 15 and NT‐proBNP as Blood‐Based Markers of Vascular Brain Injury and Dementia. Journal of the American Heart Association 2020, 9, e014659, doi:doi:10.1161/JAHA.119.014659.
- In the discussion section, the authors should be clearer and incisive mainly when (if pertinent) it is needed to compare their work with other authors/papers. To be clear when they referred to ESRD or CKD
ANS) Modifications have been reflected in the text.
Cognitive dysfunction is common in patients with ESRD CKD undergoing dialysis, and cognitive dysfunction has been reported in up to 38% of hemodialysis patients. [10,27,28]. Additionally, cognitive dysfunction in patients with decreased renal function affects not only the quality of life but also mortality. [29]. Cognitive dysfunction in CKD is thought to be caused by vascular cognitive impairment, such as stroke and transient ischemic attack, and neurodegenerative processes, such as chronic HTN, chronic cerebral inflammation, uremic toxins, and high α-amyloid levels. Malnutri-tion-inflammation-atherosclerosis syndrome and inflammation persisted in ESRD CKD patients undergoing dialysis, which may affect systemic blood vessels and organs.[30-32].
GDF-15 was reported as associated with cognitive impairment and dementia in general populations.[15,36]. In patients with CKD who are not undergoing dialysis, the eGFR shows a tendency to decrease as GDF-15 levels increase.[18,37]. Additionally, higher levels of serum GDF-15 have been reported as associated with a rapid decline in renal function.[38]. However, information regarding the relationship between cognitive function and GDF-15 in ESRD is lacking. Although GDF-15 levels are elevated in hemodialysis patients patients undergoing hemodialysis compared with patients with normal renal function, it is unclear whether serum level of GDF-15 can be used as an indicator of cognitive dysfunction in hemodialysis patients.[21].
- In the Limitations section, the authors could add that it is a limitation to use only the MMSE to assess the cognitive dysfunction.
ANS) Additional modifications were made in the limitation section.
*Limiatation part
Third, as this study was retrospective, various tools could not be used to evaluate cognitive function, and only the K-MMSE was utilized. In future research, it is believed that richer results will be obtained by employing multiple cognitive function assessment tools in retrospective studies. Forth, Among the patients included in this study, there were no individuals diagnosed with depression related to cognitive impairment.
- Check for acronyms in the text: for instance, ESRD, CDK, HTN, MTT...
ANS)HTN has been modified to hypertnesion, and the abbreviation has been re-attached.
**Result section
DM was the predominant cause of ESRD in both groups, but the rates of hypertension and glomerulonephritis differed. DM was the underlying disease in 66.1% and 60.6% and hypertension was the underlying disease in 74.6% and 87.9% of patients in groups A and B, respectively. Regarding educational background, 83% of patients in group A had received >7 years of education, whereas 48.5% of patients in group B had a low level of education
Table 1. / Table 4.
SD, standard deviation; K-MMSE, Korean Mini Mental Status Examination; ESRD, end-stage renal disease; DM, diabetes mellitus; GN, glomerulonephritis; PCKD, polycystic kidney disease; IHD, ischemic heart disease; CI, cerebral infarction ; HD, hemodialysis; BMI, body mass index; GDF 15, growth and differentiation factor 15; BUN, blood urea nitrogen; CRP, C-reactive protein; B2-MG, beta-2 microglobulin. * Student’s t-test † Chi-square tes
- Check for English errors or typos, for instance “in another few studies…” (line 56 page 2); “in in vitro..” (line 115 page 3); serum creatinine (r =.258, P = 0.0213), il P=0.013 (line 211, page 6); “to be due to decreased renal function …” (line 342 page 12).
ANS) The modifications were made in the limitation section.
*line 56 page 2
In another few studies, brain imaging showed changes associated with brain atrophy in the group with high serum GDF-15 level, and the risk of cognitive dysfunction was reportedly increased. In a few studies, brain imaging revealed changes associated with brain atrophy in the group with high serum GDF-15 levels, and the risk of cognitive dysfunction was reportedly increased [19].
*“in in vitro..” (line 115 page 3);
To confirm that similar results were obtained in in vitro and in vivo experiments regarding the results of the analysis of GDF-15 measurement and cognitive decline in hemodialysis patients, experiments were also conducted in mice.
*serum creatinine (r =.258, P = 0.0213), il P=0.013 (line 211, page 6); I have divided the sentences and corrected the typos.Serum GDF-15 (r = −.337, P = 0.001), age (r = −.435, P = 0.000), and CRP (r = −.257, P = 0.014) were negatively correlated with the K-MMSE score, and serum albumin (r =.259, P = 0.013).,sSerum creatinine (r =.258, P = 0.0213), potassium (r =.257, P = 0.0213), and phosphorus (r =.348, P = 0.001) were positively correlated with the K-MMSE score (Table 2).
“to be due to decreased renal function …” (line 342 page 12).
These findings are presumed to result from decreased renal function, in addition to age-related changes, in hemodialysis patients. These findings are presumed to be due to decreased renal function in addition to age-related changes in hemodialysis patients.
"Thank you for your comments. I have carefully reviewed the feedback provided in the comments and made the necessary revisions to the document. If you have any further questions or concerns, please feel free to let me know."